# Indian Medicinal Herbs and Formulations for Alzheimer’s Disease, from Traditional Knowledge to Scientific Assessment

**DOI:** 10.3390/brainsci10120964

**Published:** 2020-12-10

**Authors:** Jogender Mehla, Pooja Gupta, Monika Pahuja, Deepti Diwan, Diksha Diksha

**Affiliations:** 1Department of Neurological Surgery, Washington University School of Medicine, St. Louis, MO 63110, USA; diwand@wustl.edu; 2Department of Pharmacology, All India Institute of Medical Sciences, New Delhi 110029, India; ddiksha896@gmail.com; 3Division of Basic Medical Sciences, Indian Council of Medical Research, Ministry of Health and Family Welfare, Government of India, V. Ramalingaswamy Bhawan, New Delhi 110029, India; pmonika@icmr.gov.in

**Keywords:** Alzheimer’s disease, cognitive impairment, herbal medicine, memory, complimentary and alternative medicine

## Abstract

Cognitive impairment, associated with ageing, stress, hypertension and various neurodegenerative disorders including Parkinson’s disease and epilepsy, is a major health issue. The present review focuses on Alzheimer’s disease (AD), since it is the most important cause of cognitive impairment. It is characterized by progressive memory loss, language deficits, depression, agitation, mood disturbances and psychosis. Although the hallmarks of AD are cholinergic dysfunction, β-amyloid plaques and neurofibrillary tangle formation, it is also associated with derangement of other neurotransmitters, elevated levels of advanced glycation end products, oxidative damage, neuroinflammation, genetic and environmental factors. On one hand, this complex etiopathology makes a response to commonly used drugs such as donepezil, rivastigmine, galantamine and memantine less predictable and often unsatisfactory. On the other hand, it supports the use of herbal medicines due to their nonspecific antioxidant and anti-inflammatory activity and specific cholinesterase inhibitory activity. The popularity of herbal medicines is also increasing due to their perceived effectiveness, safety and affordability. In the present article, the experimental and clinical evidence have been reviewed for various Indian herbal medicines such as *Centella asiatica*, *Bacopa monnieri*, *Curcuma longa*, *Clitoria ternatea*, *Withania somnifera*, *Celastrus paniculatus*, *Evolvulus alsinoides*, *Desmodium gangeticum*, *Eclipta alba*, *Moringa oleifera* and *Convolvulus pluricaulis,* which have shown potential in cognitive impairment. Some commonly available herbal formulations for memory impairment in India have also been reviewed.

## 1. Introduction

Ayurveda mentions three aspects of mental abilities, i.e., Dhi (process of acquisition/learning), Dhuti (process of retention) and Smriti (process of recall) [1]. A dysfunction in the process of acquisition/learning, retention or recall is known as dementia. Worldwide, about 40 million elderly are living with dementia [2,3]. In India, an estimated 3.7 million elderly people have dementia, and the prevalence is expected to increase two-fold by 2030 and three-fold by 2050 [4]. Dementia is associated with neurodegenerative disorders such as Alzheimer’s disease (AD), Parkinson’s disease and epilepsy. This review is focused on the potential of herbal medicine in AD since it is responsible for more than two-thirds of all dementia cases [5,6].

Cognitive functions that are mainly affected in AD patients include memory, executive functioning, language, visuospatial functioning and attention. Several hypotheses have been proposed for establishing the cause of AD. Cholinergic hypothesis, which is the oldest theory, describes acetylcholine (ACh) deficiency as the causative factor [7]. Currently available therapies for AD management are based on this hypothesis [8]. The β-amyloid hypothesis, most cogent hypothesis [9,10,11,12] provides the basis for development of new therapeutic strategies for AD treatment [13]. The histopathological hallmarks of AD are neuritic plaque and neurofibrillary tangle (NFT) formation in the brain [14]. Other associated factors that may also contribute to neurodegeneration in AD are elevated levels of advanced glycation end products, oxidative damage and neuroinflammation (Figure 1). The involvement of free radicals and inflammation in pathogenesis of AD hint towards the possible role of antioxidant and anti-inflammatory agents as therapeutic tools [15]. Studies have also reported that antioxidants protect against Aβ induced neuronal toxicity [16,17]. Fuzhisan (FZS), a herbal drug, demonstrated a neuroprotective effect by inhibiting Aβ (25–35)-induced activation of cyclin-dependent kinase 5, calcium influx, calpain activation and tau hyperphosphorylation [18]. Inhibitory effect of an aqueous extract of *Ceylon cinnamon (C. zeylanicum)* on tau aggregation and filament formation has also been reported [19].

## 2. Limitations of Currently Approved Cognition Enhancers

Currently approved drugs for AD include cholinesterase inhibitors (donepezil, rivastigmine and galantamine) and NMDA receptor antagonist (memantine). Cholinesterase inhibitors provide only symptomatic relief of behavioral deficits without modifying the complex pathologies in mild to moderate AD patients [20]. However, memantine is mainly recommended for moderate to severe AD cases [20]. Cholinesterase inhibitors significantly improve the cognition in patients with mild to moderate AD but their efficacy for neuropsychiatric symptoms is still questionable. AD patients receiving cholinesterase inhibitors experience adverse effects like nausea, vomiting, diarrhea, dizziness, etc. [21]. The common adverse effects associated with AChE inhibitors are nausea, vomiting, diarrhea, abdominal pain, loss of appetite and weight, though these can be minimized by slow dose escalation and administration with food. Other adverse effects of AChE inhibitors such as extrapyramidal symptoms, sleep disorder and cardiorespiratory adverse effects, are associated with central cholinergic over-activity whereas muscle cramps, weakness and urinary incontinence, are associated with peripheral cholinergic over-activity [22]. These adverse effects are often dose limiting and disabling in nature. Further, AChE inhibitors do not address neuronal degeneration and associated changes in the brain.

Cholinergic dysfunction, amyloid-*β* neurotoxicity, oxidative damage and inflammation have been targeted for treatment of AD but with limited success [23,24]. Studies indicate that antioxidants (vitamins E and C) and non-steroidal anti-inflammatory drugs slow the progression of AD [25,26,27,28,29,30]. Hormone replacement therapy has also been tried as a therapeutic strategy. Though it performed better than tacrine [31], it is no longer recommended, as it may increase the risk of adverse cardiovascular events and breast cancer [32].

Memantine showed promising anti-Alzheimer effects in preclinical experimental models, however, in clinical studies, it has not shown clear therapeutic efficacy in AD [33]. It is currently being used in the treatment of moderate to severe AD. The rate of decline in behavioral and functional impairment in patients with moderate to severe AD is reduced by memantine [34]. Patients taking memantine experience adverse effects like fatigue, pain, confusion, urinary incontinence, urinary tract infection, peripheral edema, etc. [35]. Thus, the search continues for effective and affordable medicines, which when prescribed for long duration, have acceptable adverse effects or interaction with food and drugs and delay the progression or reverse the disease process.

Herbal medicines, supported by a wealth of traditional knowledge, may serve the purpose as they can target AD pathophysiology at multiple sites, both at cellular and molecular levels. Though the mechanisms of action of herbal medicines are not clear, it has been proposed that they exert their protective effects against cognitive impairment through nonspecific antioxidant and anti-inflammatory activities and through specific action on AChE, β-amyloid fibril formation and tau aggregation (Figure 2) [19,36].

Therefore, the present article reviews selected herbal drugs and formulations commonly studied for the treatment of AD.

## 3. Herbal Medicines for Alzheimer’s Disease, Experimental and Clinical Evidence

Herbal drugs and complementary medicines have been used since ancient times for treatment of neurological disorders. Several herbal medicines worldwide have been used for neurodegenerative disorders. For example, *Salvia lavandulaefolia* (Spanish sage) and *Salvia officinalis* (common sage) are being used for improving memory in Europe since the 16th century [37]; and are also supported by clinical trials [38,39]. *Bacopa monniera* (water hyssop) has been used in the Indian Ayurvedic system to improve memory and intellectual functions as an immemorial custom. *Centella asiatica* (Asiatic pennywort), another Ayurvedic remedy, is given in combination with milk to improve memory [40]. *Withania somnifera* root, a rejuvenative tonic, is also used in Ayurveda to enhance memory [41,42]. Herbal medicines are becoming popular due to their perceived effectiveness, safety and affordability. Indeed, only recently, scientific studies have started providing evidence and support for the use of herbal medicines in memory related disorders.

Various CNS active Indian herbal medicines like *Withania somnifera*, *Centella asiatica*, *Celastrus paniculatus* and *Bacopa monnieri* have shown cognitive improvement in experimental models of AD when given as prophylactic treatment [43,44,45,46,47,48]. A randomized, double-blind exploratory trial reported comparable efficacy of a *Gingko biloba* extract and donepezil in AD patients with associated neuropsychiatric problems. The combination was reported to be superior to donepezil monotherapy in terms of both safety and efficacy [49].

### 3.1. Centella asiatica

Plant description: *Centella asiatica* (*C. asiatica*), a small, annual herb belonging to the family Apiceae is found throughout India and commonly known as mandukparni or jalbrahmi. It has small fan-shaped green leaves with white or light purple-to-pink or white flowers and it bears small oval fruit [50]. The leaves of mandukparni have been used as a memory enhancer in the Ayurvedic system of medicine [51]. Its use has also been described in the African system of medicine, and traditional Chinese medicine. It is used to delay ageing, prevent memory related disorders and is given with milk to enhance memory [40].

Main chemical constituents: The main chemical constituents of *C. asiatica* are asiaticosides, asiatic acid, madecassoside and madasiatic acid [50]. Other chemical compounds isolated from *C. asiatica* are brahmoside and brahminoside, isothankuniside, thankuniside and centelloside [50].

Pharmacological activities: *C. asiatica* is well known for its broad pharmacological activities such anti-inflammatory, antioxidative stress, antiapoptotic effects, neuroprotective effects, wound healing, antipsoriatic, antiulcer, hepatoprotective, antidepressant activity, nootropic activity, anticonvulsant, sedative, immunostimulant, cardioprotective, antidiabetic, cytotoxic and antitumor, antiviral, antibacterial, insecticidal and antifungal [50].

Preclinical studies: Aqueous extract of *C. asiatica* in 100, 200 and 300 mg/kg doses given orally for 14 days has been reported to dose-dependently improve cognitive functions in normal rats [52]. Pretreatment with the extract for 21 days significantly reversed streptozotocin induced cognitive impairment [51]. The authors attributed the beneficial effect of *C. asiatica* to antioxidant activity as evidenced by a decrease in malondialdehyde, increase in glutathione, catalase and superoxide dismutase levels. A study by Rao et al. [53] demonstrated that 15 days treatment with *C. asiatica* at a dose of 200 mg/kg from day 15 to 30 postpartum stimulated learning and memory in rats, which lasted for at least 6 months postpartum. They also observed an increase in dendritic arborization of hippocampal CA3 neurons, which may be one reason for improvement in brain function. Another study showed improved cognitive outcome in elderly subjects following prescribed dose of 500 mg/b.i.d dried *C. asiatica* for a 6-month period [54]. Dhanasekaran et al. [55] found that an 8 month treatment with 2.5 mg/kg of aqueous extract of *C. asiatica* significantly decreased amyloid beta 1-40 and 1-42 levels in the hippocampus of PSAPP transgenic mice expressing “Swedish” amyloid precursor protein and M146L presenilin 1 mutations, which result in spontaneous amyloid beta plaque formation. A reduction in Congo red stained fibrillar amyloid plaques was detected on the long-term treatment with 5.0 mg/kg dose.

*C. asiatica* aqueous leaf extract showed improvement in learning and memory in rats, and modulated dopamine, 5-hydroxytryptamine (5-HT) and noradrenaline systems in the rat brain in-vivo [56]. The leaf extract also had sedative, antidepressant and cholinomimetic activities [57] suggesting its suitability for treatment of AD associated cognitive dysfunction and depression and anxiety. The leaf extract stimulated dendrites of neuronal cells in the rat brain [51] and induced neurite elongation in human SH-SY-5Y cells and accelerated axonal regenerate in rats [58]. Cyclic AMP response element binding property (CREB) and its phosphorylated form are involved in memory formation [59]. Reduced level of phosphorylated CREB has been reported in AD patients and experimental models of AD [60]. The aqueous extract of *C. asiatica* leaves enhanced phosphorylation of CREB in both neuroblastoma cells, which express inducible Aß and in cortical primary cells, which were chronically exposed to external Aß in-vitro. The extract increased neuronal dendritic arborization and axonal regeneration in rats [51,58,61].

Triterpenoids are the major active component of ethanolic extract of *C. asiatica*, which consists of many chemical constituents such as asiatic acid, mecadessic acid, asiaticoside, scentellin, asiaticin and centellicin [62,63,64]. Asiatic acid and its derivatives have shown a promising memory improving effect [65] by improving ACh synthesis [66,67]. It has been patented (Hoechst Aktiengesellschaft) for the treatment of dementia and as a cognition enhancer. The exact constituent responsible for cognition enhancing effects of the herb remains to be established. However, studies suggest that perhaps triterpene saponins present in the leaf improve cognitive function by influencing central neurotransmitters.

Clinical evidence: In a randomized, double-blind placebo-controlled, study, *C. asiatica* extract was administered to healthy volunteers as 250–750 mg once daily dose for 2 months. The high dose enhanced working memory and improved self-rated mood [68].

Thus, clinical and experimental studies support memory enhancing potential of *C. asiatica.* However, its use for treatment of AD remains to be evaluated.

Toxicity: *C. asiatica* extract and asiaticoside were found to be well tolerated in experimental studies. Asiaticoside did not cause any toxicity up to 1 g/kg oral dose [69]. In acute toxicity study, *C. asiatica* extract up to 10 g/kg did not shown any sign of toxicity whereas in the subacute toxicity study, no toxicity was observed when the extract was administered at the doses of 10–1000 mg/kg. In the chronic toxicity study, doses up to 1200 mg/kg/day for six months did not result in significant toxicity in Wistar rats [70]. However, in one study, oral administration of 1000 mg/kg/day dried *C. asiatica* for 30 days caused hepatotoxicity in albino rats [71].

### 3.2. Bacopa monnieri

Plant description: *Bacopa monnieri* (*B. monniera*)*,* belonging to the Scrophulariaceae family is a small, perennial creeping herb with numerous branches, small oblong leaves and light purple or white flowers. In India, it is commonly called Brahmi and is known for its revitalizing, Medhya rasayana and nootropic activities as it strengthens memory and intellect (Medhya). Bacopa has been used for the treatment of various ailments for thousands of years by the practitioners of the traditional system of medicine of India [72].

Main chemical constituents: The main chemical compounds of *B. monniera* are triterpenoid saponins known as bacosides. The alkaloids brahmine, nicotine and herpestine have also been reported in this plant. Novel saponins called bacopasides I–XII have also been identified [72].

Pharmacological activities: This medicinal herb possesses various biological activities such as anticonvulsant, antidepressant, anxiolytic, analgesic, anti-inflammatory, antioxidant, antimicrobial, antiulcerogenic, anti-*Helicobacter pylori*, adaptogenic, antineoplastic, bronchodilatory, hepatoprotective and immunostimulatory [72].

Preclinical studies: The extract of *B. monniera* has been reported to contain several beneficial bioactive components such as alkaloids, flavonoids, glycoside, triterpenoids saponins and alcohols. The alcoholic extract of *B. monniera* improved acquisition, consolidation and retention of memory in the foot shock motivated brightness discrimination test, active conditioned avoidance test and Sidman continuous avoidance responses in rats [73,74]. Bacosides A and B (a mixture of 2 saponins) may be responsible for its facilitatory effect on learning and memory. Besides, bacosides has been proven for its antioxidant and anti-inflammatory effects [75] bacosides also attenuated the retrograde amnesia produced by immobilization induced stress, electroconvulsive shock and scopolamine [76]. They enhanced protein kinase activity and increased the protein content in the hippocampus, which may also contribute to their memory enhancing effect [74,77,78]. Administration of bacosides (200 mg/kg) for 3 months in middle-aged and aged rats exerted a protective effect against age associated alterations in the neurotransmission system, behavioral paradigms, hippocampal neuronal loss and oxidative stress markers [79]. The involvement of the microRNA 124-CREB pathway and serotonergic receptor in the memory enhancing mechanism of standardized extract of *B. monniera* (BESEB CDRI-08) has also been reported [80,81].

The effect of alcoholic extract of *Bacopa* has been evaluated at the dose of 20, 40 and 80 mg/kg on cognitive functions and neurodegeneration in the animal model of AD induced by bilateral intracerebroventricular administration of AF64A. They found that *Bacopa* improved the escape latency in the Morris water maze test and prevented the reduction in cholinergic neuron density [47,82]. Besides, oral administration of 40 mg/kg/day of the *Bacopa* extract for 5 weeks prevented neurotoxicity in rats exposed to aluminum chloride [83]. Cognitive deficit induced by intracerebroventricular (ICV) injection of cholchicine and ibotenic acid into the nucleus basalis magnocellularis was attenuated by standardized *Bacopa* extract by reversing the depletion of ACh level, reduction in choline acetyl transferase (ChAT) activity and decrease in muscarinic cholinergic receptor binding in frontal cortex and hippocampus [84]. Holcomb et al. [43] reported that administration of ethanolic extract of *Bacopa* leaves at doses of 40 and 160 mg/kg for 2 and 8 months reduced A*β* 1–40 and 1–42 levels in the cortex of PSAPP mice. Bacopa, at the dose of 50 mg/kg, demonstrated the neuroprotective effect in the colchicine model of dementia through its antioxidant effect and restored the activity of Na^+^K^+^ATPase and AChE [85]. The neuronal dendritic growth stimulating property of *Bacopa* has also been reported which may be responsible for its memory enhancing property [86].

Clinical evidence: In a double-blind, placebo-controlled trial in 38 healthy volunteers (ages 18–60 years), single dose of 300 mg *B. monniera* extract (containing 55% combined bacosides A and B) did not cause any significant change in cognitive function at 2 h [87]. However, six week *Bacopa* administration (300 mg for subjects under 90 kg, and 450 mg for subjects over 90 kg, equivalent to 6 g and 9 g dried rhizome, respectively) in a double-blind, randomized, placebo controlled fashion was associated with significant improvement in retention of new information in 40–65 year old healthy adults. Though there was no difference in the rate of acquisition of information [88].

Stough et al. [89] reported significant improvement in verbal learning, memory consolidation and speed of early information processing following *Bacopa* administration (containing 55% combined bacosides) for 12 weeks at a dose of 300 mg daily in a double-blind placebo-controlled study in healthy volunteers (age 18–60 years, *n* = 46). Since the effects were not observed until five weeks of treatment, the slow onset of action may be attributed to *Bacopa*’s antioxidant properties and/or its effect on the cholinergic system. In another randomized, double-blind, placebo-controlled trial in 54 elderly participants without clinical signs of dementia (mean age 73.5 years), similar *Bacopa* treatment enhanced an auditory verbal learning test, delayed word recall memory scores and a stroop test relative to the placebo [14]. In subjects above 55 years of age with memory impairment, standardized *Bacopa* extract 125 mg was given twice daily for 12 weeks in a double blind, placebo-controlled manner. There was a significant improvement in mental control, logical memory and paired associated learning [90]. Furthermore, *Bacopa* extract at the dose of 300 mg/kg, daily for 12 weeks improved memory acquisition and retention in healthy older Australians population [91].

In children (age 6–8 years), *Bacopa* syrup (350 mg *Bacopa* powder), when administered three times a day for three months, resulted in significant improvement as compared to the placebo [92]. However, this study was not blinded. Negi et al. [93] carried out a double-blind, randomized, placebo-controlled trial in 36 children diagnosed with attention deficit/hyperactivity disorder (mean age 8.3–9.3 years). Nineteen children received *Bacopa* extract (standardized to contain 20% bacosides) at a dosage of 50 mg twice daily for 12 weeks. As compared to placebo, a significant improvement in cognitive function was observed in *Bacopa*-treated children at 12 weeks as evidenced by improvement in sentence repetition, logical memory and paired associate learning tasks, which was maintained at 16 weeks (after four weeks of placebo administration).

Toxicity: The LD_50_ of orally administered *Bacopa* extracts in rats was 5 g/kg for aqueous extract and 17 g/kg of the alcoholic extract [77]. The intraperitoneal LD_50_ was 1000 mg/kg for aqueous extract and 15 g/kg for alcoholic extract [94]. A double-blind, placebo-controlled trial in healthy male volunteers reported safety and tolerability of bacosides in single (20–30 mg) and multiple (100–200 mg) daily doses over a four-week period [77]. A randomized, double-blind, placebo-controlled trial reported that *Bacopa* treatment (300 mg/kg, daily) for 12 weeks caused increased stool frequency, abdominal cramps and nausea, which may be due to either an upregulation of ACh level or saponin-mediated gastrointestinal tract irritation, or both [91].

### 3.3. Curcuma longa

Plant description: *Curcuma longa* (*C. longa*) Linn is a perennial herb belonging to the family Zingiberaceae. It is grown for commercial use in South and Southeast Asia. Curcumin, also known as turmeric, is obtained from the rhizome of the plant, and is commonly used in India as a food flavoring and coloring agent. Several preparations of the plant have been used for centuries in the Ayurvedic system of medicine [95].

Main chemical constituents: Curcuminoids are main chemical constituents of turmeric, which include mainly curcumin (diferuloyl methane), demethoxycurcumin and bisdemethoxycurcmin. Other chemical compounds reported in this plant are alpha- and beta-tumerone, artumerone, alpha- and gamma-atlantone, curlone, zingiberene and curcumol [96].

Pharmacological activities: Previous studies reported the various pharmacological properties of curcuminoids such as neuroprotective, analgesic, antiproliferative, anti-inflammatory, anticancer, antidiabetic, hypocholesterolemic, antithrombotic, antihepatotoxic, antidiarrheal, carminative, diuretic, antirheumatic, hypotensive, antimicrobial, antiviral, antioxidant, larvicidal, insecticidal, antivenomous and antityrosinase effects [97].

Preclinical studies: It is also one of the most systematically studied plants for various diseases [98]. It has been reported in various experimental studies to possess wide variety of biological and pharmacological activities including antioxidant, anti-inflammatory and cholesterol-lowering properties, all three of which are key processes involved in the pathogenesis of AD.

Water insolubility is a major limitation for curcumin, which has been overcome, to some extent, by synthesis of biodegradable poly (lactic-co-glycolic acid) (PLGA) coated curcumin nanoparticles. These nanoparticles were found to be able to destroy amyloid aggregation and exhibit antioxidative activity without a cytotoxic effect [99,100]. Nanoliposomes of curcumin have high affinity for Aβ1-42 fibrils and were found to inhibit the formation of fibrillar and oligomeric Aβ in-vitro [101,102]. Apolipoprotein E3 mediated poly(butyl) cyanoacrylate nanoparticles containing curcumin (ApoE3-C-PBCA) provided photostability, enhanced the cellular uptake of curcumin and increased its efficacy against Aβ induced cytotoxicity [103]. Curcumin also demonstrated a protective effect against Aβ neurotoxicity by decreasing Aβ production through downregulation of presenilin 1 (PS1) and GSK-3-β expression and accelerating Aβ fibril conversion [104,105].

Curcumin has been shown to reduce both in-vivo and in-vitro Aβ plaque deposition [106,107]. Curcumin treatment for six months significantly decreased the elevated levels of oxidized protein and proinflammatory interleukin-1β in the transgenic APPSw mouse brain (Tg2576) [106]. Plaque formation and the concentration of insoluble and soluble Aβ were also lowered by curcumin in the same study. Pretreatment with curcumin (10, 20 and 50 mg/kg, p.o for 21 days) ameliorated memory impairment in the sporadic AD model in mice [108]. Furthermore, curcumin in diet form improved the spatial memory, oxidative stress and synaptophysin loss via reducing Aβ deposits [109]. Significant cognitive improvement was documented at low (160 ppm) and high (1000 ppm) doses of curcumin after administration for the 6-month period in the double transgenic AD model (APP/PS1) [110]. In-vivo, curcumin may protect cells from beta amyloid attack and subsequent oxidative stress-induced damage [111]. Curcumin can inhibit Aβ aggregation or promote its disaggregation at low concentrations (IC_50_ = 0.81–1 μM). Monomeric Aβ formed fewer aggregates in the presence of curcumin, whereas increasing doses of curcumin promoted disassembly of preformed Aβ aggregates. Structurally, curcumin is similar to Congo red and can prevent oligomer formation after binding to plaques and recognize secondary structure in fibrillar and oligomeric Aβ. Low dose curcumin significantly lowered the soluble Aβ levels, insoluble amyloid and plaque burden by nearly 40% [106]. Additionally, curcumin treatment for 7 days caused reduction in plaques burden and reversed structural changes in dystrophic dendrites in APPswe/PS1dE9 mouse model of AD [112].

Impaired insulin or insulin-like growth factor-1 (IGF-1) signaling is associated with AD. It leads to hyperphosphorylation of the tau protein, mitochondrial dysfunction, oxidative stress and necrosis, and contributes to cognitive impairment [113,114,115]. Curcumin significantly improved cognitive function by improving the IGF-1 level in the intracerebroventricular (ICV)-streptozotocin (STZ) model of sporadic AD [116]. It also suppressed IL-1 and glial fibrillary acidic protein, reduced oxidative damage and plaque burden and decreased the amount of insoluble amyloid [26]. Another experimental study showed that curcumin treatment restored learning and memory functions in the STZ model of AD by reducing the oxidative stress, enhancing ChAT activity and restoring insulin receptor protein [117,118].

Curcumin suppressed the microgliosis in neuronal layers, but it failed to reduce within plaques microgliosis and even significantly increased microgliosis immediately adjacent to plaques, raising the possibility that it may stimulate microglial phagocytosis of amyloid. Other possible mechanisms for curcumin induced neuroprotective effects include inhibition of IL-1-induced increase in alpha-1-antichymotrypsin (α_1_ACT) and NFκB-mediated transcription of apolipoprotein E (ApoE). Both α_1_ACT [119,120] and ApoE [121,122,123,124] have been shown to be proamyloidogenic in APP transgenic mice. Curcumin can also reduce two other proamyloidogenic factors, oxidative damage [125,126] and raised cholesterol levels [127]. The neuroprotective effect of curcuminoid mixture and its individual components on inflammatory and apoptotic gene expression in AD using an Aβ plus ibotenic acid-infused rat model has also been reported [128]. Additionally, Ahmed and colleagues also reported that a curcuminoids mixture (bisdemethoxycurcumin, demethoxycurcumin and curcumin) treatment improved memory function in amyloid fragment induced AD-like conditions in rats [129]. Nonetheless, chronic treatment with curcumin also prevented the colchicine induced cognitive impairment in rats by reducing the oxidative stress [130].

Chronic stress induces impairment of spatial cognition, neuroendocrine and plasticity abnormalities due to an increase in serum corticosterone levels. Curcumin exerts its neuroprotective effect by normalizing the corticosterone response, resulting in downregulating of calcium/calmodulin kinase II and glutamate receptor (NMDA-2B) levels [131]. The protective effect of curcumin on a Aβ1–40 AD model was documented by Wang et al. [132] and Yin et al. [133], where treatment with 300 mg/kg curcumin reversed spatial learning and memory impairment accompanied by hippocampal regeneration. Evidence also suggests that metals are concentrated in the AD brain and curcumin chelates iron and copper (but not zinc) bound to beta amyloid potentially contributing to amyloid reduction [134]. A different approach was followed by McClure et al. [135], where aerosol-mediated treatment of young 5XFAD mice with curcumin averted Aβ buildup and memory deficits in adulthood as compared to the untreated mice.

Thus, this multitarget compound is a promising therapeutic agent for AD and associated cognitive decline. However, despite intensive curcumin related research in various diseases, there is a lack of clinical data on the efficacy of curcumin in AD.

Toxicity: In a phase I trial with 25 healthy subjects, curcumin up to 8000 mg/day for 3 months did not show any toxicity [136]. In an acute toxicity study, ethanolic extract of rhizome of *C. longa* at the doses of 0.5, 1.0 and 3.0 mg/kg did not cause any sign of toxicity in mice. Moreover, no toxicity was found at 100 mg/kg/day in the 90-day toxicity study in mice [137].

### 3.4. Clitoria ternatea

Plant description: *Clitoria ternatea* (*C. ternatea*) is a perennial tropical climber herb with slender downy stem, found throughout the tropical regions of India, growing wild and in gardens, bearing white or blue flowers. *C. ternatea* belongs to family Fabaceae commonly called “butterfly”. It is a commonly used Ayurvedic medicine. *C. ternatea* is called Aparajit (Hindi), Aparajita (Bengali) and Kakkattan in Indian traditional medicine [138]. The extracts of *C. ternatea* have been used in Ayurveda, as an ingredient in “Medhya rasayana”.

Main chemical constituents: Various phytocomponents such as taraxerol, teraxerone, ternatins, delphinidin-3, delphinidin-3ß-glucoside, malvidin-3ß-glucoside, 3 monoglucoside, 3-rutinoside, 3-neohisperidoside, 3-o-rhamnosyl Glycoside, kaempferol-3-o-rhamnosyl, aparajitin, beta-sitosterol, malvidin-3ß-glucoside, kaemphferol, p-coumaric acid, etc., are isolated from *C. ternatea* [138].

Pharmacological activities: In previous studies, various biological activities including nootropic, anticonvulsant, antidepressant, antianxiety, antistress, antioxidant, anti-inflammatory, antihyperlipidemic, antidiabetic, antiasthmatic, analgesic, immunomodulatory, cytotoxicity, platelet aggregation inhibitory, antimicrobial, gastroprotective and hepatoprotective of *C ternatea* have been documented [138].

Preclinical studies: The nootropic activity of methanolic extract of aerial parts of *C. ternatea* (100 mg/kg, p.o) has been reported by using elevated plus maze and the object recognition test in rats [139]. Taranalli and Cheeramkuzhy evaluated the ethanolic extracts of roots and aerial parts of *C. ternatea* at the dose of 300 and 500 mg/kg, p.o in amnesia induced by submaximal electroshock [140]. They also estimated the ACh level in the whole brain and different parts of it. The aerial parts extract resulted in improved memory retention and increased brain ACh content, which was more at 300 mg/kg as compared to the 500 mg/kg dose. The root extract exhibited similar but more marked effects, which were almost equal at both doses.

Rai et al. [141] described the learning and memory enhancing effect of the *C. ternatea* root extract during the growth spurt period in rats. They intubated 7-day old neonatal rats and administered 50 and 100 mg/kg of the aqueous root extract of *C. ternatea* for 30 days. The extract improved retention in the passive avoidance task and spatial performance in the T-maze test. The behavioral changes were reported to be long lasting as indicated by a 30 days post-treatment evaluation. A previous study also showed that the aqueous root extract (50 and 100 mg/kg, p.o for 30 days) enhanced dendritic arborization of amygdala neurons in rats [142]. This cognition enhancing effect was hypothesized to be due to the presence of growth factors similar to the brain derived neurotrophic factor or nerve growth factor. Increase in hippocampus acetylcholine content [139] may be one of the reasons for nootropic activity of *C. ternatea* root. In addition, Rai [143] reported that the *C. ternatea* root extract exhibited the neurogenesis-promoting sequel on the anterior subventricular zone of neural stem cells. More recently, Damodaran et al. [144] documented the neuroprotective effect of the *C. ternatea* root extract in reversing chronic cerebral hypoperfusion-induced neural damage and memory impairment at doses of 200 and 300 mg/kg. In another study, Mehla and colleagues showed anti-AD effects of *C. ternatea* in ICV-STZ induced AD-like conditions in rats [145]. These observations suggest that *C. ternatea* extract exerts its beneficial effect by preventing the progression of cognitive deterioration in AD. However, the potential of *C. ternatea* extract still needs to be systematically evaluated for human use.

Toxicity: Ethanolic extract of aerials parts and root of *C. ternatea* have been studied at 200–3000 mg/kg, p.o in mice. A cathartic effect of root extract was observed. Mice treated with a dose above 2000 mg/kg had ptosis and were lethargic. The extract was not lethal orally but resulted in severe CNS depression and death when used intraperitoneally at dose of 2900 mg/kg and above [140]. Taur and Patil [146] reported LD_50_ of ethanolic extract of *C. ternatea* root to be more than 1300 mg/kg.

### 3.5. Withania somnifera

Plant description: *Withania somnifera* (*W. somnifera*) is a small woody shrub belonging to the family Solanaceae and is widely grown in India. It is commonly called Indian ginseng or winter cherry or ashwagandha. Its flowers are greenish or yellowish in color and about one centimeter long [147,148]. Ashwagandha is mentioned in ancient Sanskrit writings from India as a “Medhya rasayan”. It is also known as Indian ginseng and is widely used in Ayurveda. It is an ingredient in many formulations prescribed as a general tonic to increase energy, improve overall health and longevity [147,148].

Main chemical constituents: The major phytoconstituents of *W. somnifera* are isopellertierine, anferine, withanolides, withaferins, sitoindoside VII and VIII and withanoloides. Other chemical compounds are withanine, somniferine, somnine, somniferinine, withananine, pseudo-withanine, tropine, pseudo-tropine, 3-a-gloyloxytropane, choline and cuscohygrine [149,150,151,152].

Pharmacological activities: *W. somnifera* exhibits a broad range of biological activities like anti-inflammatory, antioxidant, neuroprotective, antischemic, anti-Parkinson’s, antiepileptic, anxiolytic, antidepression, antiarthritic, cardioprotective, antidiabetic, anticancer, antistress, nephroprotective, heptoprotective, antihypoxic, immunomodulatory, hypolipidaemic and antimicrobial [152].

Preclinical studies: Total alkaloid extract (ashwagandholine, AG) of *W. somnifera* root has been studied for its effects on CNS [153]. *W. somnifera* attenuated the memory loss induced by STZ through the antioxidant mechanism [154]. The root preparation has been shown to have protective effects in neurodegenerative disorders by reducing stress induced degeneration in the brain hippocampus of rats [155]. The extract containing sitoindosides VII–X and withaferin A (50 mg/kg, p.o for two weeks) reversed ibotenic acid-induced cognitive deficit and reduction in cholinergic markers (e.g., ACh and ChAT) in rats [156]. Sitoindosides VII-X and withaferin differentially (40 mg/kg for 7 days) but favorably altered the AChE activity and enhanced M_1_- and M_2_-muscarinic receptor-binding in various brain regions [157]. Withaferin A and Withanolide A suggested to have a potent immunomodulatory effect in BV-2 microglial cells by triggering the Nrf2 pathway, leading to production of the neuroprotective protein, such as heme oxygenase-1 [158].

Withanoside IV, another chemical constituent of *Withania*, when administered orally at the dose of 10 micromol/kg prevented cognitive impairment in the experimental model of AD [44]. Sominone (1 microM) a metabolite of Withanoside IV, induced axonal and dendritic regeneration and synaptic reconstruction in cultures of rat cortical neurons damaged by the amyloid peptide, Aβ(25–35) [44]. Therefore, withanoside IV may act as a prodrug, with sominone as the active component. The enhancement of spatial memory by sominone may be attributed to neuritic outgrowth, which is mediated by the neurotrophic factor receptor, RET [159]. Methanolic root extract dose dependently enhanced in-vitro dendrite formation in human neuroblastoma cells [159]. A study carried out by Jayaprakasam et al. [160] stated that withanamides (A/C) present in *W. somnifera* fruits protect pheochromocytoma-(PC-12) from β-amyloid induced toxicity. In the same study, β-amyloid fibril formation was prevented, possibly due to the presence of a serotonin moiety in both withanamide compounds.

Treatment with *Withania* root extract (1 g/kg, p.o for 30 days) reversed the AD pathology by upregulating the low-density lipoprotein receptor-related protein, which enhanced the Aβ clearance and ameliorated the cognitive deficit in middle-aged and old APP/PS1 mice [161]. Alcoholic extract of the *Withania* leaf and its component withanone was neuroprotective against scopalmine induced changes in the brain [162]. An in-vitro, inhibitory effect on the fibril formation by Aβ peptide has also been reported [163]. The increase in cortical muscarinic ACh receptor capacity might partly explain the cognition-enhancing and memory-improving effects of *Withania*. The root extract and their chemical constituents such as glycowithanolides also possess anxiolytic, antidepressant, anti-inflammatory and antioxidant activities, which may be relevant in AD treatment [164,165]. Furthermore, withanone, a chemical constituent from root extract of *W. somnifera* showed improvement in cognitive functions by inhibiting amyloid processing and reducing the elevated levels of proinflammatory cytokines and oxidative stress markers [166]. *W. somnifera* (20 mg/mL) treatment mitigated the Aβ toxicity and mediated longevity in the AD model of *Drosophila melanogaster* [167].

Clinical evidence: A prospective, randomized, double-blind, placebo-controlled study reported that treatment with ashwagandha-root extract (300 mg twice daily for eight weeks) improved immediate and general memory functions and enhanced executive function, attention and information processing speed in adults with a mild cognitive impairment [168]. In a systematic review, Ng and colleagues mentioned that *W. somnifera* extract ameliorated cognitive impairment and improved executive functions in adults with mild cognitive impairment [169]. There is limited data available on the clinical use of *Withania* for cognitive impairment.

Toxicity: Different preparations and extracts of *W. somnifera* root did not cause any toxicity even on chronic treatment [170]. Ashwagandholine 2% suspension in propylene glycol had a LD_50_ of 465 mg/kg in rats and 432 mg/kg in mice [171]. Whereas intraperitoneal administration of aqueous-methanol root extract caused 50% lethality in mice at a dose of 1076 ± 78 mg/kg [172]. Equimolar combination of sitoindosides VII and VIII and withaferin-A (SG-2) when administered once intraperitoneal, the LD_50_ was 1564 ± 92 mg/kg [172].

### 3.6. Celastrus paniculatus

Plant description: *Celastrus paniculatus* (*C. paniculatus*) is a large climber of the family Celastraceae. It grows throughout India, on sub-Himalayan slopes and the hilly regions of Punjab and South India. It is commonly known as jyotismati, which comes from the Sanskrit words “jyoti teja” or fire of mind and “mati”—intelligence. Traditionally, the bark and seeds have been used as a brain tonic, to promote intellect and to improve digestion, stimulant and expectorant [173]. In Ayurveda, *C. paniculatus* has been used to treat many diseases like depression, leprosy, paralysis, fever and arthritis. The seed oil and fruit are commonly used for their tranquilizer, sedative and wound healing properties [174].

Main chemical constituents: *C. paniculatus* shows the presence of various phytoconstituents such as sesquiterpenoid polyalcohols and esters (malkanguniol, malkangunin, polyalcohol A–D and celapnin); alkaloids (paniculatine and celastrine); phenolic triterpenoids (celastrol and paniculatadiol); fatty acids (oleic, linoleic, linolenic, palmitic, stearic and lignoceric acid) and agarofuran derivatives [175].

Pharmacological activities: Various pharmacological activities such as hypolipidemic, neuroprotective, anti-infertility, antiarthritic, wound healing, anti-inflammatory, antioxidant, analgesic, antimalarial, antibacterial and fungicidal action of *C. paniculatus* have been reported [176].

Preclinical studies: Celastrus seed extract and oil have been evaluated in different experimental models of cognitive impairment such as scopolamine and sodium nitrite induced amnesia. The aqueous, methanolic, chloroform and petroleum ether extracts of seeds of *C. paniculatus* were investigated for their effect on cognitive function in rats. The aqueous extract showed significant improvement in cognitive performance at the doses of 200 and 300 mg/kg, p.o for 14 days. In another study, methanolic extract reported to have memory-enhancing activity in rats at doses of 100, 200 and 400 mg/kg [177]. The antioxidant activity of *C. paniculatus* may be involved in improving the cognitive function [45]. The oil of *C. paniculatus* seeds when given for 14 days to Wistar rats at a dose of 400 mg/kg resulted in enhanced learning and memory in radial arm maze and decreased the AChE enzyme activity in hypothalamus, frontal cortex and hippocampus [178]. Karanth et al. [179] also demonstrated a similar effect of *C. paniculatus* at the dose of 400 mg/kg for 3 days. In another study, rats treated with 850 mg/kg of *C. paniculatus* oil for 15 days had significantly improved retention in two passive avoidance tasks [56]. The seed oil treatment for 14 days at the doses of 50, 200 and 400 mg/kg, p.o reversed scopolamine induced spatial memory impairment in the Morris water maze and increased locomotor activity without affecting AChE activity in rats [180]. The aqueous seed extract improved memory performance in elevated plus maze and in sodium nitrite induced amnesia by reducing the AChE activity [181]. Furthermore, *C. paniculatus* seed oil treatment showed memory improvement in scopolamine induced amnesia in mice [182]. *C. paniculatus* has not undergone clinical trials for safety and efficacy. Animal toxicology data is also lacking to date.

### 3.7. Evolvulus alsinoides

Plant description: *Evolvulus alsinoides* L. (*E. alsionoides,* dwarf morning glory), belonging to the family Convolvulaceae, is a perennial herb with small woody and branched rootstock. *E. alsionoides* is a weed, found mainly in the swampy regions of tropical and subtropical regions of the world. It has numerous branches (greater than 30 cm) with long hairs. The leaves are small, acute, elliptical with small size and blue-colored flowers [183]. It is locally known as Shankhpushpi and is very commonly used in Ayurveda. It is a key ingredient in majority of Medhya Rasayana formulations available in the Indian market. It is traditionally used as a memory enhancer in children and elderly and for neurological disorders like epilepsy [184].

Main chemical constituents: Major chemical constituents are octadecanoic acid, n-hexadecanoic acid, piperine, squalene, ethyl oleate and cholesterol [185].

Pharmacological activities: Studies indicate that *Evolvulus alsionoides* (*E. alsionoides*) possesses in-vitro antioxidant [186], immunomodulatory [187], adaptogenic, antiamnesic [188] and antiulcer [189] activities.

Preclinical studies: Nahata et al. [190] reported learning and memory enhancing property of its ethanolic extract and ethyl acetate and aqueous fractions in rats. The ethanolic extract (100 mg/kg, p.o) also protected against scopolamine induced dementia in rats [188]. Three days oral treatment with *E. alsionoides* (100 mg/kg) was effective in decreasing scopolamine induced deficit in adult male Swiss mice [188]. Pretreatment with hydro-alcoholic extract at the doses of 100, 300 and 500 mg/kg, p.o ameliorated the ICV-STZ induced cognitive impairment by decreasing the oxidative stress and rho kinase (ROCK II) expression in the rat brain [15,145]. In-vitro, aqueous and hydroalcoholic extracts of *E. alsinoides* showed free radicals scavenging, anti-inflammatory and enzymes (cholinesterase, glycogen synthase kinase-3-β, Rho kinase (ROCKI I), prolyl endopeptidase, catechol-o-methyl transferase and monoglycerol lipase) inhibitory activity, all of which are involved in the pathophysiology of AD [15]. Previous studies also indicated the memory enhancing effect of *E. alsionoides* in the experimental model of amnesia [191,192]. The methanol and water extract of *E. alsinoides* documented to exhibit acetylcholinesterase activity, supporting its potential in reverting neuronal dysfunctions and thus in management of AD [193]. *E. alsionoides* has not been studied systematically for clinical efficacy and toxicological effects.

### 3.8. Desmodium gangeticum

Plant description: *Desmodium gangeticum* (*D. gangeticum*)*,* belonging to the family Fabaceae, commonly known as Salpani in Hindi and is found in abundance throughout India. It is a perennial undershrub, 60–130 cm high with somewhat angular branches. Its leaves are simple, ovateoblong or rounded with purplish or white flowers, 4–7 cm [194]. It has been used in the traditional system of medicine as a bitter tonic, febrifuge, antiemetic, digestive and in various inflammatory conditions due to vata disorder [195]. In Satpuda hills of India, powdered root of *D. gangeticum* is applied along with honey to treat a mouth ulcer. In Uttat Pradesh state of India, the leaf paste of *D. gangeticum* and aloe vera are applied to prevent hair fall [196].

Main chemical constituents: *D. gangeticum* shows the presence of alkaloids (tryptamines and phenylethylamines), pterocarpanoids (gangetin and desmodin), phospholipids, sterols, flavone and glycosides [197].

Pharmacological activities: It shows various pharmacological activities including antileishmanial, immunomodulatory, antioxidant, anti-inflammatory, antinociceptive, cardioprotective, antiulcer, antiamnesic and hepatoprotective [194].

Preclinical studies: Aqueous extract of *D. gangeticum* when administered orally at the dose of 50, 100 and 200 mg/kg for 7 days improved memory in mice [198,199]. Scopolamine and ageing induced amnesias were also prevented in rats by pretreatment with the aqueous extract of *D. gangeticum* [198]. Moreover, treatment of mice with the chloroform extract (400 mg/kg) and alkaloidal fraction (50 mg/kg) of *D. gangeticum* for 6 days alleviated the scopolamine-induced amnesia [200]. Antioxidant, anti-inflammatory and AChE inhibitory activity of *D. gangeticum* has also been reported [199,201,202]. These pharmacological properties indicate the potential of *D. gangeticum* in the management of AD related cognitive impairment. Yet, not much clinical evidence is available to this effect. Toxicity studies are also required to establish the safety of this potentially useful herb.

### 3.9. Eclipta Species

Plant description: *Eclipta alba* (L.) Hassk (*E. alba*) is an annual erect or prostrate herb, belonging to the Asteraceae family. There are four major varieties of Eclipta based on the colors of flower like red, yellow, white and blue. The flowers of *E. alba* are white in color and largely harvested due to its therapeutic activity [203]. Its stem is reddish-purple in color with up-turned hairs and roots are greyish with cylindrical shape [204]. *Eclipta alba* (*E. alba*), commonly known as Bringharaj, is well known in the traditional system of medicine for its beneficial effects on learning and memory [205]. Another species of Eclipta, commonly known as false daisy, is *E. prostrate*. It has also been traditionally used for treatment of memory related disorders, hepatic disorders and atherosclerosis [206].

Main chemical constituents: The major chemical constituents present in *E. alba* are coumestans, flavonoids, sterols, alkaloids, triterpenoid saponins and volatile oil.

Pharmacological activities: It has good antimicrobial properties like antibacterial, antifungal and antimalarial. It also shows antidiabetic, hepatoprotective, hypolipidemic, anticancer, hair growth promoting and memory enhancement and immunomodulatory properties [207].

Preclinical studies: The ethanolic extract of *E. alba* resulted in improvement in learning and memory abilities in passive avoidance and the elevated plus maze test in rats after both acute and chronic administration [207]. Saponins, the main chemical constituent of butanol fraction of *E. prostrate*, prevented ethanol induced memory impairment in rats [208]. Kim et al. [209] also reported that butanol fraction increased ACh content, decreased MAO-B activity and reduced oxidative stress in the rat brain. Lipid lowering and antioxidant activities of *Eclipta* plants have also been reported [210]. *E. alba* also possesses antiviral, antinociceptive, anti-inflammatory, bronchodilator, antibacterial, antipyretic, tonic, expectorant and hepatoprotective activity [211,212]. Previous study also reported the improvement in learning and memory functions of rats [213]. Based on the animal data available, the herb needs to be evaluated clinically.

Toxicity: An aqueous extract of *E. alba* did not cause any toxicity at a dose of 2.0 g/kg orally and 200 mg/kg by intravenous and intraperitoneal routes. The LD_50_ in mice were 7.841 g/kg, 302.8 and 328.3 mg/kg for oral and intravenous and intraperitoneal routes respectively [137]. The alcoholic extract did not show any toxicity in rats and mice and the minimum lethal dose was found to be greater than 2.0 g/kg when given orally and intraperitonially in mice [214].

### 3.10. Moringa oleifera

Plant description: *Moringa oleifera* (*M. oleifera*) belonging to the family Moringaceae is the commonly distributed species of this family. This plant is native to India and the height of trees can reach up to 10 m. It has fragile branches and bipinnate or tripinnate leaves. It has yellowish white flowers 0.5–1 cm long and around 2 cm broad [215]. It is commonly known as a drumstick. *M. oleifera* has shown antimicrobial activity and traditionally been used to clarify water due to its coagulant property. Oil of *M. oleifera* has high stability and contains a large amount oleic acid, hence used as an edible oil, biodiesel and lubrication of machinery [216].

Main chemical constituents: The major chemical constituents in *M. oleifera* are vitamins (vitamin A and C), polyphenols (flavonoids, chlorogenic acid and phenolic acids), alkaloids, glucosinolates, isothiocyanates, tannins and saponins [217].

Pharmacological activities: Various pharmacological activities like nootropic, anti-inflammatory, hypocholesterolemic, hypotensive and antioxidant effects of its leaves have been reported [218,219,220,221,222]. Additionally, it has also shown hypolipidemic, antiobesity, antidiabetic, anti-inflammatory, immunomodulatory and anticancer effects. *M. oleifera* is a good source of vitamin, hence prevents night-blindness and delays cataract development [217].

Preclinical studies: Pretreatment with *M. oleifera* at an oral dose of 250 mg/kg prevented hypoxia induced memory impairment in rats by maintaining the monoamines levels in the brain [223]. The ethanolic leaf extract at a dose of 250 mg/kg, p.o for 14 days provided protection against cognitive impairment induced by ICV–colchicine. It restored colchicine induced changes in the brain norepinephrine, serotonin and dopamine levels [224]. Improvement in learning and memory has been suggested to be due to its antioxidant effect. Other studies also demonstrated the protective effect of *M. oleifera* against memory impairment in experimental models of dementia [225,226]. Intriguingly, *M. oleifera* was shown to mitigate hyperphosphorylation and Aβ pathology also in hyperhomocysteinemia-induced AD in rats [227]. The mechanism of action, composition of the herb and difference between different extracts need to be established before it can be taken to clinical trials.

Toxicity: The aqueous leaf extract was found safe in rats after oral administration of 2000 mg/kg [228]. The acute toxicity of aqueous and ethanolic extract of *M. oleifera* root was evaluated in mice with the LD_50_ of 15.9 g/kg and 17.8 g/kg, respectively [229].

### 3.11. Convolvulus pluricaulis

Plant description: *Convolvulus pluricaulis* (*C. pluricaulis*) Choisy is a perennial, wild, prostrate herb, which belongs to the Convulvulaceae family and is mainly found in Northern India. It has long branches of about 30 cm and blue flowers. Its leaves are elliptical in shape and located alternately with flowers and branches [230]. It is commonly known as shankhpushpi and is used as a nervine tonic in the Ayurvedic system of medicine, to improve memory and intellect [230]. It is classified as Medhya rasayana (promotes intellectual capacity) and Majjadhatu rasayana (rejuvenates the nervous tissue). The leaves of *C. pluricaulis* have been used for depression and other mental disturbances [231].

Main chemical constituents: The major chemical components are alkaloids (shankhpushpine and convolamine), volatile oils, favanoid-kampferol, phytosterol, amino acids, fatty acids, scopoletin and beta-sitosterol (Sethiya NK) [232].

Pharmacological activities: Various neuropharmacological actions such nootropic, antistress, antidepressant, anxyiolytic, anticonvulsant and sedative activities of this plant are well reported [233,234,235,236]. Furthermore, it also possesses antiamnesic, antiulcer, anticatatonic, antibacterial, immunomodulatory and cardiovascular activity [232].

Preclinical studies: The ethanolic extract of *C. pluricaulis* and its ethyl acetate and aqueous fraction at the dose of 100 and 200 mg/kg, p.o showed memory enhancing properties in Cook and Weidley’s Pole Climbing Apparatus, passive avoidance paradigms and active avoidance tests [237]. Convolvine, a chemical constituent of *C. pluricaulis* potentiated the effect of arecoline (memory enhancer) and improved cognitive dysfunction in AD [238,239]. Sharma et al. [240] also reported that the ethanolic extract at 100 and 200 mg/kg oral dose significantly improved memory in young and aged mice but the retention was better in young mice. *C. pluricaulis* also possesses antioxidant and hypolipidemic effects, which may be partially responsible for improvement in cognitive function [190,241]. *C. pluricaulis* administration for 3 months at the dose of 150 mg/kg prevented aluminum chloride induced neurotoxicity by decreasing AChE activity, reducing oxidative stress and preserving the activity of ChAT and Nerve Growth Factor-Tyrosine kinase A receptor (NGF-TrkA) [242]. Alcoholic extract of *C. pluricaulis* Choisy (leaves) showed Aβ production inhibition in-vitro [243]. Additionally, isolated bioactive coumarins from *C. pluricaulis* ameliorated scopolamine induced amnesia in mice [244]. Despite detailed experimental studies, the herb has not been evaluated clinically.

Toxicity: *C. pluricaulis* has not been studied for toxicity. *C. microphyllus,* another plant of the same family had the LD_50_ of 1250 mg/kg after oral administration of the whole plant extract [245].

## 4. Other Plants with Potential Memory Enhancing Activity

Several other plants may improve cognitive functions and be useful in AD. However, very limited, if any, literature is present to review the plants individually. These less explored plants include *Acorus calamus* (vach), *Prunus amygdalus* (badam), *Orchis mascula* (salap), *Syzygium aromaticum* (lavang), *Mukta pishti* (pearl), *Tinospora cordifolia* (guduchi), *Picrorrhiza kurroa* (kutki), *Zingiber officinale* (sonth), *Boerhaavia diffusa* (punarnava)*, Commiphora wightii (guggal*)*, Piper longum* (pippali)*, Carum copticum* (ajwain)*, Cyperus rotundus* (coco-grass), *Santalum album* (Indian sandalwood)*, Elettaria cardamomum* (cardamom)*, Foeniculum vulgare* (fennel)*, Rosa damascene* (damask rose) and *Cinnamomum cassia* (cassia).

## 5. Methodology

Search criteria: Database searches were conducted on PUBMED, and GOOGLE SCHOLAR using keywords: dementia, herbal products/drugs/medicine, Alzheimer’s disease and complementary and alternative medicines. The searches were limited to those plant/plant products, which are mentioned in Indian Ayurvedic literature for their potential use in some form of dementia and literature available online in the English language.

Inclusion criteria: The following studies were included in the present review article: (1) preclinical studies (in-vitro and in-vivo studies); (2) clinical studies and (3) herbal medicine identified with their regional or Hindi name.

Exclusion criteria: The following criteria was used in the present review to exclude the studies: (a) herbal drugs of non-Indian origin; (b) articles available in the language other than English and (c) full text not available.

## 6. Indian Herbal Formulations Studied in Alzheimer’s Disease

### 6.1. Mentat

Compound formulations are commonly used in Ayurveda, based on the concept that such combinations provide synergistic therapeutic effect with minimal adverse effects. BR-16A (Mentat) is a polyherbal formulation used as Medhya Rasayana in Ayurveda and is used to improve memory and cognitive deficits associated with chronic illness and aging [156]. The ingredients in BR-16A are Brahmi (*Bacopa monnieri*), Mandookaparni (*Centella asiatica*), Ashwagandha (*Withania somnifera*), Shankapushpi (*Evolvulus alsinoides*), Jatamansi (*Nardostachys jatamansi*), Vach (*Acorus calamus*), Tagar (*Valeriana wallachii*), Badam (*Prunus amygdalus*), Salap (*Orchis mascula*), Lavang (*Syzgium aromaticum*), Pearl (*Mukta pishti*), Malkangni (*Celastrus paniculatus*) and Sonth (*Zingiber officinale*).

Mentat has been shown to augment acquisition and retention of learning in rats and prevented cognitive deficits induced by variety of insults including prenatal undernutrition, postnatal environmental impoverishment, sodium nitrite hypoxia, aluminum, increasing age and electroconvulsive shock induced antero-grade and retro-grade amnesia [246,247,248]. Administration for 20 days at a dose of 100 mg/kg/day significantly prolonged the shortened step-through latency induced by aluminum administration and also significantly improved retention of learning in aged rats [248]. Ramteke et al. [249] reported that administration of BR-16A facilitated learning and memory in rats on the Hebb Williams complex maze as compared to control. BR-16A also showed dose dependent improvement in learning and memory in scopolamine induced amnesia in rats [250]. Mentat when administered for 2 weeks reversed the cognitive deficit and cholinergic dysfunction induced by colchicine and ibotenic acid model of AD [251].

Clinical evidence: It improved memory quotient of normal subjects in different age groups [252], increased memory span and attenuated fluctuations of attention in normal adults and improved learning ability in children with behavioral problems or minimal brain damage [253].

Toxicity: In an acute toxicity study, Mentat did not show any sign of toxicity up to the dose of 1.5 g/kg. The LD_50_ was found to be 1.75 g/kg after intraperitoneal injection [254] and 2400 mg/kg after oral administration [250].

### 6.2. Trasina

Trasina is a polyherbal formulation of some Indian medicinal plants, which are classified as Medhya rasayana in Ayurveda. It consists of *Withania somnifera* (80 mg), *Ocimum sanctum* (190 mg), *Eclipta alba* (10 mg), *Tinospora cordifolia* (10 mg), *Picrorrhiza kurroa* (10 mg) and shilajit (20 mg). It has shown significant nootropic effect at a dose of 200 and 500 mg/kg, p.o when administered for 21 days in colchicine and ibotenic acid induced cognitive impairment. Trasina, dose dependently, improved both memory and cholinergic markers like acetylcholine concentration, choline acetyl transferase activity and muscarinic cholinergic receptor binding in the frontal cortex and hippocampus of rat brain after 14 and 21 days of treatment. Thus, its nootropic effect may be attributed to the correction of cholinergic dysfunction [255].

### 6.3. Memorin

Memorin consists of *Mandookparni* (60 mg), Shankhpushpi (60 mg), Jatamansi (30 mg), Yashtimadhu (60 mg) and Smruti sagar (60 mg). The effect of memorin was evaluated by Andrade, 1998 in an age related memory disorder and reported beneficial effects in elderly persons who experienced age related memory decline [256]. Additionally, memorin (200 mg/day/kg) was found to attenuate retrograde and anterograde amnesia in rats when tested using passive avoidance learning paradigms in the shuttle box and T-maze test [257].

### 6.4. Bramhi Ghrita

It is a polyherbal Ayurvedic formulation that contains *Bacopa monneri* (40% *w*/*w*), *Evolvulus alsinoids* (20% *w*/*w*), *Acorus calamus* (20% *w*/*w*), *Saussurea lappa* (20% *w*/*w*) and cow’s ghee (750 mL). Traditionally, it is used as a memory enhancer [258]. Achliya et al. [259] evaluated the learning and memory enhancing effect of this formulation at 30, 50 and 100 mg/kg oral doses. The results of this study showed that *Bramhi Ghrita* at the doses of 50 and 100 mg/kg, p.o decreased the transfer latency in elevated plus maze and escape latency in Morris water maze test. Additionally, it enhanced the learning and memory of rats indicating the nootropic activity [260].

### 6.5. Abana

Abana, another polyherbal Ayurvedic formulation, is available in tablet form consisting of *Terminalia arjuna* (30 mg), *Withania somnifera* (20 mg), *Nepeta hindostana* (20), Dashamoola (20 mg), *Tinospora cordifolia* (10 mg), *Phyllanthus emblica* (10 mg), *Terminalia chebula* (10 mg), *Eclipta alba* (10 mg), *Glycyrrhiza glabra* (10 mg), *Asparagus racemosus* (10 mg), *Boerhaavia diffusa* (10 mg), Shilajeet (20 mg), *Centella asiatica* (10 mg), *Convolvulus pluricaulis* (10 mg), *Ocimum sanctum* (10 mg), *Nardostachys jatamansi* (10 mg), *Piper longum* (10 mg), *Carum copticum* (10 mg), *Zingiber officinale* (10 mg), Shankh bhasma (10 mg), Makardhwaj (10 mg), *Cyperus rotundus* (5 mg), *Acorus calamus* (5 mg), *Embelia ribes* (5 mg), *Syzygium aromaticum* (5 mg), *Celastrus paniculatus* (5 mg), *Santalum album* (5 mg), *Elettaria cardamomum* (5 mg), *Foeniculum vulgare* (5 mg), *Rosa damascena* (5 mg), *Cinnamomum cassia* (5 mg), Jaharmohra (10 mg), Abhrak bhasma (5 mg), Akik pishti (5 mg), Yeshab pishti (5 mg), Yakut pishti (5 mg), Praval pishti (5 mg) and *Crocus sativus* (2 mg).

Abana was administered for 15 days at the doses of 50, 100 and 200 mg/kg orally to young and aged mice and retention memory was tested using elevated plus the maze and passive avoidance test. It was also tested in scopolamine and diazepam induced amnesia at same doses. Abana reduced the brain AChE activity in a dose dependent manner. The results of these studies indicate that Abana improves memory, which may be due to reduction in brain AChE activity. Acute oral administration of Abana in mice did not cause any toxicity up to the dose of 2000 mg/kg [261].

## 7. Herbal Drugs: Regulatory Status

The regulatory guidelines for herbal medicines differ from country to country. USFDA classifies herbal medicines into dietary supplements and botanical drugs. Safety and efficacy studies are not needed for marketing of dietary supplements, but they should be so labeled. For botanicals, description of the product and documentation of prior human experience of the product is required. The requirements may vary from non-clinical studies to clinical trials, batch effect analysis [262]. In European Union, Committee on Herbal Medicinal Products (HMPC) issues scientific opinions on herbal substances and preparations [263]. The regulatory pathways depend on prior human exposure and range from traditional use registration; well established use marketing authorization and to stand alone or mixed application.

In India, herbal medicines are governed by Drugs and Cosmetics Act 1940, and Rules 1946. The development pathway is similar to other synthetic drugs, if they have to be incorporated into the modern system of medicine as per new drugs and clinical trials rule, 2019 [264]. Licensing, composition, formulation and manufacturing of products, labeling, packing and quality is done as per Schedule T [265]. Safety and efficacy studies are undertaken in accordance with AYUSH GCP guidelines [266].

## 8. Issues and Challenges with Herbal Drugs

Quality control of herbal drug: Extraction technique and processing step may cause variation in the concentration of active constituents, which necessitates quality control of herbal medicines. The macroscopic and microscopic property of herbal medicines should be examined for quality control. Determination of ash value, heavy metals, pesticide residues and microbial contamination should be carried out.

Herb-drug interaction: Coadministration of herbal drugs with prescribed drugs may result in serious adverse effects. Herbal drugs contain numerous unidentified constituents, which make it difficult to assess the nature of interactions. Additionally, it is generally believed that herbal medicines are safe since they belong to a natural origin but recently many of the herbs were found to exhibit adverse drug reactions [267]. Some reports also showed that adverse events are caused due to the herb–drug interaction [267,268,269]. Heterogeneity in doses and frequency of use also obstruct precise assessment of drug interactions. CYP450 is involved in the metabolism of drugs used for management of AD [270,271]. Hence CYP450 inhibition by herbal drugs should be assessed for predicting potential herb–drug interactions. *Ginkgo biloba* when given with donepezil cause an increased effect in AD due to additive cholinergic activity. However, when it was given with phenytoin, it causes breakthrough seizures due to the induction of CYP2C19. Curcumin increase the oral bioavailability of celiprolol due to inhibition of intestinal CYP450 enzymes and p-glycoprotein [272]. Moreover, coadministration of curcumin and donepezil (reversible cholinesterase inhibitor) had a synergistic effect on cognition and oxidative stress [273] and good BBB permeability [274]. More experimental and clinical studies need to be performed to evaluate the herb–drug interaction. Such interactions may be prevented with disclosure of concomitant use by the patients and awareness of physicians. In the elderly population, the absorption, metabolism and elimination of drugs are already impaired. Concomitantly use of herbal medicines may worsen the impairment. Therefore, herbal drugs should be used cautiously in elderly patients.

Adulteration: Herbal drugs are many times substituted or adulterated with other inferior products with morphological resemblance of authentic herb. This type of adulteration is more common for herbs with volatile components. The adulterants may not have a therapeutic benefit or may even cause adverse effects. Therefore, quality assurance of herbal medicines should be mandatory.

Labeling of herbal medicines: Proper labeling can reduce the risk of inappropriate use and adverse effects. The label should contain the name and amount of herbal drug and active ingredients, direction for intake, its intended use, storage conditions, shelf life, adverse effects and warnings, if any.

Pharmacovigilance for herbal medicines: Modern times rely more on the systematically studied modern medicines as they must adhere to stringent national and international regulations also. On the other hand, traditional systems of medicine suffer from a lack of or inadequate regulatory guidelines. In addition, herbal medicines are widely perceived to be safe due to their natural origins. However, as reviewed above, several herbal medicines exhibit adverse effects on their own or due to adverse herb–drug interactions with concomitant medicines. Apart from these inherent risks of herbal medicines, several adverse effects associated with them may be due to improper labeling, unknown composition, a lack of standardization, inferior quality, contamination, adulteration, improper use and even quackery.

Patients with Alzheimer’s disease constitute a special subgroup due to their vulnerability and their inability to communicate adverse events, in later stages of the disease. This population needs special attention with respect to the monitoring of adverse effects and drug–drug interactions. Hence, there is a need to integrate pharmacovigilance of herbal medicines with that of the modern medicine, under the national pharmacovigilance programs.

## 9. Conclusions and Future Prospectus

The alternative systems of medicine have been used since ancient times and different extracts of medicinal plants and herbal formulations have demonstrated potential for use in AD. Medicinal plants provide a fertile ground for new drug discovery because of the presence of various chemical constituents and their ability to act on different biological targets. However, much work remains to be done to translate this potential into actual medicine. Standardization of plant extracts is an urgent need in herbal drug research. Phytoconstituents responsible for pharmacological activities should be isolated, identified and systematically tested. Multicenter clinical trials should be performed to validate the efficacy of these herbal medicines alone or in the form of formulations for the treatment of AD. The present article reviewed the reported efficacy of herbal medicines against AD in experimental and clinical studies.

## Figures and Tables

**Figure 1 brainsci-10-00964-f001:**
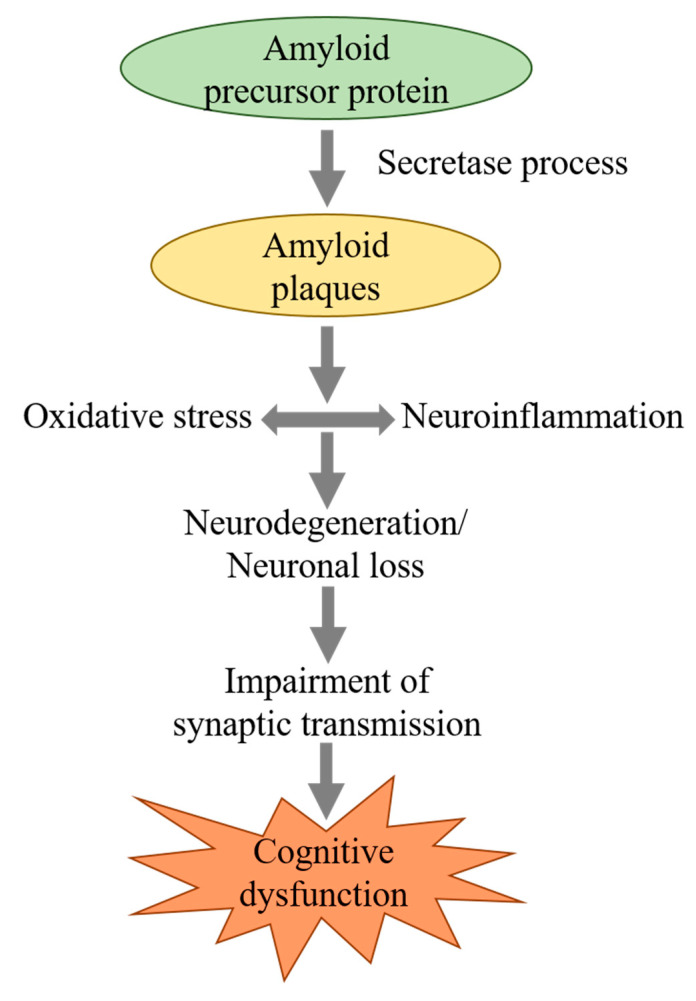
General pathogenesis of Alzheimer’s disease (APP, amyloid precursor protein).

**Figure 2 brainsci-10-00964-f002:**
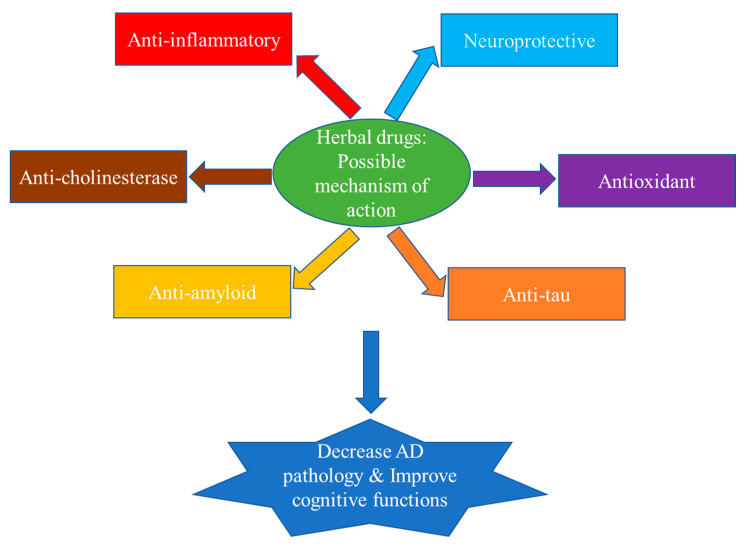
Multipronged approach of herbal medicines in Alzheimer’s disease.

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
