# Peer review of "Indian Medicinal Herbs and Formulations for Alzheimer’s Disease, from Traditional Knowledge to Scientific Assessment"

_brainsci, 2020, doi:10.3390/brainsci10120964_

Round 1

Reviewer 1 Report

  1. Kindly remove tacrine (it has been removed from the market in several countries due to severe toxicity).
  2. Figure-1: The colors are overwhelming and not clear
  3. Kindly validate this statement with appropriate reference (s)- “Memantine showed promising anti-Alzheimer effects in preclinical experimental models, 83 however, in clinical studies, it has shown no therapeutic efficacy in AD.”
  4. Figure-2: What do your mean by “neuroprotective”? It looks redundant
  5. Kindly look into the Table-1 again. There have been clinical trials for Bacopa
  6. The authors have reviewed the neuroprotective effects of
    1. various plants “Centella asiatica, Bacopa monnieri, Curcuma longa, Clitoria ternatea, Withania somnifera, Celastrus paniculatus, Evolvulus alsinoides, Desmodium gangeticum, Eclipta species, Moringa oleifera, Convolvulus pluricaulis, other plants.
    2. Various formulations: Mentat, Trasina, Memorin, Bramhi Ghrita, Abana.
  7. The authors have also reviewed the adverse effects.

Author Response

Comments and Suggestions for Authors

Comment: Kindly remove tacrine (it has been removed from the market in several countries due to severe toxicity).

Reply: We thank the reviewer for useful suggestion. As per suggestion, we have removed tacrine from the text.

Comment: Figure-1: The colors are overwhelming and not clear.

Reply: We have modified the figure and corrected figure-1 is included in the revised manuscript.

Comment: Kindly validate this statement with appropriate reference (s)- “Memantine showed promising anti-Alzheimer effects in preclinical experimental models, 83 however, in clinical studies, it has shown no therapeutic efficacy in AD.”

Reply: We thank the reviewer for useful comment. Following reference has been added in the revised manuscript, for the above-mentioned statement. The addition is highlighted in the revised manuscript.

Folch J, Busquets O, Ettcheto M, Sánchez-López E, Castro-Torres RD, Verdaguer E, Maria Luisa Garcia ML, Olloquequi J, Casadesús G, Beas-Zarate C, Pelegri C, Vilaplana J, Auladell C,  Camins A. Memantine for the Treatment of Dementia: A Review on its Current and Future Applications. J Alzheimers Dis. 2018; 62;1223–1240.

Comment: Figure-2: What do your mean by “neuroprotective”? It looks redundant.

Reply: Several herbal drugs including curcumin, Withania somnifera, Centella asiatica prevented the neuronal damage in various experimental models. Therefore, we mentioned “Neuroprotection” one of the several mechanism. 

Comment: Kindly look into the Table-1 again. There have been clinical trials for Bacopa.

Reply: The clinical trials for bacopa have been included in the text. To avoid confusion, we have removed the table in the revised manuscript.

Reviewer 2 Report

The Review paper of Mehla et al. examines in great detail the treatment potential of selected herbal drugs for Alzheimer’s disease (AD). While the text if of interest, and I together with the authors do believe the herbal medicines are of great interest for development of new drugs, I raise a number of issues, which in my opinion are not addressed adequately by the text.

Introduction:

Line 82-83 – the sentence about memantine needs citation(s).

The authors describe the current treatment options for AD. I suggest they also mention current guidelines.

Last paragraph (lines 96-98) reads: “Therefore, the present article reviews selected herbal drugs and formulations commonly used for memory related disorders. Some of the pre-clinical and clinical studies of plant extracts relevant to AD are mentioned in table 1.”

I have several remarks regarding that:

1 . “reviews selected herbal drugs” – selected how?

2. “formulations commonly used” – where? How the authors selected the formulations?

3. “Some of the pre-clinical and clinical studies…” – some? Why no all available or all relevant or all meeting certain standards?

These issues should be explained.

Methods:

This section is completely missing. While it does not have to be a separate section (may be in the end of intro or in other suitable place), I believe, the authors should provide information about databases, key-words, selection and exclusion criteria, etc. Ideally the whole selection algorithm.

Another issue is the focus on AD only instead of on all types of dementia. The authors report preclinical data, which in great part are not specific for AD. The clinical trials appear to be mostly AD related. This should be explained.

Table 1: Tables in review papers are a great way to summarize the available evidence. However, to my eyes, this table does not provide much information. For preclinical studies I miss information on type of animal model, performed memory (or other) test, sex of the subject, age (where appropriate). For clinical studies I miss the number of patients, type of the study (case study?, double blind randomized one?). For both types I miss dosing and the type of extract or other formulation. Including all this information would expand the table greatly, but the table would be very meaningful. I suggest the authors consider to following options: 1) as along as everything is already described in the text, maybe a table is not necessary; 2) how about splitting the table to two, one preclinical and one clinical; 3) maybe only the clinical data as the most relevant require a table.

Section 3. Herbal medicines for AD – I am not sure I understand the point of this text section. Perhaps it belongs to the introduction?

4.x sections (the description of the specific herbs) – this text is obviously going in great depth of the matter and I like it. I would just organize it in a more standard way, ideally into paragraphs appearing consistently in all sub-sections. For instance: plant description – active compounds – preclinical data – clinical data – toxicity (to mimic what the authors generally do, but not everywhere consistently). Please, unify the texts and if there is missing evidence, say so. Consider including a section describing basic pharmacologic properties (pharmacodynamics, pharmacokinetics…) and interaction potential if known. (I appreciate, that you touched the subject in the last section of the MS.) There is a nice review about ADME in herbal medicines (He at al. 2011, ADME Properties of Herbal Medicines in Humans: Evidence, Challenges and Strategies, doi: 10.2174/138161211795164194), just for inspiration.

Line 251 – reference to figure 3. I did not find this figure.

Please, specify if the extract was standardized (and how) or not, where possible.

Section 6 (herbal formulations) – please indicate what is the regulatory status of these products, where are available, etc. If possible, please, include doses of the herbal material in the mixtures.

Author Response

Comments and Suggestions for Authors

The Review paper of Mehla et al. examines in great detail the treatment potential of selected herbal drugs for Alzheimer’s disease (AD). While the text if of interest, and I together with the authors do believe the herbal medicines are of great interest for development of new drugs, I raise a number of issues, which in my opinion are not addressed adequately by the text.

Introduction:

Comment: Line 82-83 – the sentence about memantine needs citation(s).

Reply: Following reference for the line 82-83 has been incorporated in the revised manuscript. The addition is highlighted in the revised manuscript.

Folch J, Busquets O, Ettcheto M, Sánchez-López E, Castro-Torres RD, Verdaguer E, Maria Luisa Garcia ML, Olloquequi J, Casadesús G, Beas-Zarate C, Pelegri C, Vilaplana J, Auladell C,  Camins A. Memantine for the Treatment of Dementia: A Review on its Current and Future Applications. J Alzheimers Dis. 2018; 62;1223–1240.

Comment: The authors describe the current treatment options for AD. I suggest they also mention current guidelines.

Reply: Thank you for bringing up this important point, but we would like to reiterate the fact that the present article discusses the herbal treatment of AD whereas majority of the guidelines deals with the management of AD as per modern medicine.

Comment: Last paragraph (lines 96-98) reads: “Therefore, the present article reviews selected herbal drugs and formulations commonly used for memory related disorders. Some of the pre-clinical and clinical studies of plant extracts relevant to AD are mentioned in table 1.”

I have several remarks regarding that:

1 . “reviews selected herbal drugs” – selected how?

  1. “formulations commonly used” – where? How the authors selected the formulations?
  2. “Some of the pre-clinical and clinical studies…” – some? Why no all available or all relevant or all meeting certain standards?

These issues should be explained.

Reply: In the current review article, commonly used herbal drugs or formulations for the management of memory related disorders mentioned in Indian Traditional system of medicine have been reviewed.

Additionally, we reviewed all the pre-clinical and clinical studies of plant extracts relevant to AD in the present review article. To avoid confusion, we have removed the table in the revised manuscript.

Methods:

Comment: This section is completely missing. While it does not have to be a separate section (may be in the end of intro or in other suitable place), I believe, the authors should provide information about databases, key-words, selection and exclusion criteria, etc. Ideally the whole selection algorithm. Another issue is the focus on AD only instead of on all types of dementia. The authors report preclinical data, which in great part are not specific for AD. The clinical trials appear to be mostly AD related. This should be explained.

Reply: We thank the reviewer for raining important issues. As this is not a systematic review article, therefore, we have not used any inclusion or exclusion criteria for selection of the herbal drugs.

Additionally, we appreciate useful suggestion to include all type of dementia. However, it’s very difficult to cover all type dementia as dementia is very broad topic. Considering the length of article, we just focused on AD so that we could covered all relevant information about herbal drugs and AD. 

Comment: Table 1: Tables in review papers are a great way to summarize the available evidence. However, to my eyes, this table does not provide much information. For preclinical studies I miss information on type of animal model, performed memory (or other) test, sex of the subject, age (where appropriate). For clinical studies I miss the number of patients, type of the study (case study?, double blind randomized one?). For both types I miss dosing and the type of extract or other formulation. Including all this information would expand the table greatly, but the table would be very meaningful. I suggest the authors consider to following options: 1) as along as everything is already described in the text, maybe a table is not necessary; 2) how about splitting the table to two, one preclinical and one clinical; 3) maybe only the clinical data as the most relevant require a table.

Reply: The suggestions made by reviewer can be easily found in the text. As per reviewer, table only provide basic information. Therefore, to avoid confusion between text and table, we have removed the table in the revised manuscript.

Comment: Section 3. Herbal medicines for AD – I am not sure I understand the point of this text section. Perhaps it belongs to the introduction?

Reply: We appreciate the reviewer for useful point. Now, revised sections can be found in the manuscript.

Comment: 4.x sections (the description of the specific herbs) – this text is obviously going in great depth of the matter and I like it. I would just organize it in a more standard way, ideally into paragraphs appearing consistently in all sub-sections. For instance: plant description – active compounds – preclinical data – clinical data – toxicity (to mimic what the authors generally do, but not everywhere consistently). Please, unify the texts and if there is missing evidence, say so. Consider including a section describing basic pharmacologic properties (pharmacodynamics, pharmacokinetics…) and interaction potential if known. (I appreciate, that you touched the subject in the last section of the MS.) There is a nice review about ADME in herbal medicines (He at al. 2011, ADME Properties of Herbal Medicines in Humans: Evidence, Challenges and Strategies, doi: 10.2174/138161211795164194), just for inspiration.

Reply: We appreciate valuable suggestion made by reviewer. However, in present article, we mainly focused to review the beneficial effects of herbal medicines reported in preclinical and clinical studies. Therefore, we think, there is no need to include the sections such as plant description – active compounds, basic pharmacologic properties (pharmacodynamics, pharmacokinetics). Additionally, herb-drug interactions have been covered under the heading “Issues and challenges with herbal drugs”.

Comment: Line 251 – reference to figure 3. I did not find this figure.

Reply: We apologize for it. Actually, Line 251 references to figure 2. There is no figure 3 in the article. We feel sorry for the inconvenience and have rectified the error.   

Comment: Please, specify if the extract was standardized (and how) or not, where possible.

Section 6 (herbal formulations) – please indicate what is the regulatory status of these products, where are available, etc. If possible, please, include doses of the herbal material in the mixtures.

Reply: We have incorporated the doses of herbal material in the mixture. These additions are highlighted in the revised manuscript.

Following para regarding “Regulatory status of herbal drugs” has been added in the revised manuscript. These additions are highlighted in the revised manuscript.

Herbal Drugs: Regulatory Status: The regulatory guidelines for herbal medicines differ from country to country. USFDA classifies herbal medicines into dietary supplements and botanical drugs. Safety and efficacy studies are not needed for marketing of dietary supplements, but they should be so labelled. For botanicals, description of product and documentation of prior human experience of the product is required. The requirements may vary from non-clinical studies to clinical trials, batch effect analysis (USFDA, 2016). In European Union, Committee on Herbal Medicinal Products (HMPC) issues scientific opinions on herbal substances and preparations (EMA). The regulatory pathways depend on prior human exposure and range from traditional use registration; well established use marketing authorization; to stand alone or mixed application.

In India, herbal medicines are governed by Drugs and Cosmetics Act 1940, and Rules 1946. Development pathway are similar to other synthetic drug, if they have to be incorporated into modern system of medicine as per new drugs and clinical trials rule, 2019 (Ministry of Health and Family Welfare). Licensing, composition, formulation and manufacturing of products, labelling, packing, quality is done as per Schedule T (Schedule T). Safety and efficacy studies are undertaken in accordance with AYUSH GCP guidelines (Department of AYUSH).

REFERENCES

  1. Botanical Drug Development: Guidance for Industry. 2016
  2. Human Regulatory- Herbal medicinal products. https://www.ema.europa.eu/en/human-regulatory/herbal-medicinal-products
  3. Ministry of Health and Family Welfare. New Drugs and Clinical Trials Rule, 2019. p. 149.
  4. Schedule T: Good manufacturing practices for Ayurvedic, Siddha and Unani Medicines.
  5. Department of AYUSH. Good Clinical Trial Practices for Clinical Trials in Ayurveda, Siddha and Unani medicine (GCP-ASU). 2013

Reviewer 3 Report

This review summarizes the experimental and clinical evidence for various Indian herbal formulations for improving memory impairment, notably cognitive function of Alzheimer disease (AD). The authors select a good view of this area and list the related studies' results. Moreover, they reported a detailed restatement of previous findings with its own brief summary. Table 1 presented the core information and conceptual frame of this review. Limitations of the study are stated clearly in section 7.

Accordingly, I recommend this manuscript for publication if the authors adequately respond to my concerns itemized below. Please carefully check and amend the manuscript.

  • In section 1. Introduction: “Ayurveda mentions three aspects of mental abilities…” It is recommended to quote the relative citation to enhance clarity and readability to the readers.
  • Please add a paragraph to clarify your method of inclusion and exclusion of articles for the review. A flowchart indicating the experimental design might be helpful.
  • Some formatting in the table needs to be corrected. For instance: Experimental… Besides, please also indicate the quality of the studies provided.
  • While narrative review (NR) articles are great papers to read to keep us up to date, please carefully check descriptions of main text to avoid any subjective opinions. Western medicines are paramount in the circumstance obviously. Conclusion of the paper should compare the pros and cons of Western and herbal medicines in the treatment of AD. From this study, what can be recommended for the future clinical practice against AD?

Author Response

Comments and Suggestions for Authors

This review summarizes the experimental and clinical evidence for various Indian herbal formulations for improving memory impairment, notably cognitive function of Alzheimer disease (AD). The authors select a good view of this area and list the related studies' results. Moreover, they reported a detailed restatement of previous findings with its own brief summary. Table 1 presented the core information and conceptual frame of this review. Limitations of the study are stated clearly in section 7.

Accordingly, I recommend this manuscript for publication if the authors adequately respond to my concerns itemized below. Please carefully check and amend the manuscript.

Comment: In section 1. Introduction: “Ayurveda mentions three aspects of mental abilities…” It is recommended to quote the relative citation to enhance clarity and readability to the readers.

Reply: As per the reviewer’s suggestion, following reference has been incorporated for above mentioned statement. The addition is highlighted in the revised manuscript.

Dua JS, Prasad DN, Tripathi AC, Gupta R. Role of traditional medicine in neuropsychopharmacology. Asian J Pharm Clin Res. 2009;2,72-76.

Comment: Please add a paragraph to clarify your method of inclusion and exclusion of articles for the review. A flowchart indicating the experimental design might be helpful.

Reply: We thank the reviewer for raising important issues. As this is not a systematic review article, we have not used any inclusion or exclusion criteria for selection of the herbal drugs. Based on our research experience with herbal drugs and Alzheimer’s disease, we have reviewed the commonly used herbal drugs or formulations for the management of memory related disorders mentioned in Indian Traditional system of medicine.

Comment: Some formatting in the table needs to be corrected. For instance: Experimental… Besides, please also indicate the quality of the studies provided.

Reply: To avoid confusion text and table, we have removed the table in the revised manuscript.  

Comment: While narrative review (NR) articles are great papers to read to keep us up to date, please carefully check descriptions of main text to avoid any subjective opinions. Western medicines are paramount in the circumstance obviously. Conclusion of the paper should compare the pros and cons of Western and herbal medicines in the treatment of AD. From this study, what can be recommended for the future clinical practice against AD?

Reply: We appreciate the reviewer’s valuable suggestion. We already discussed the limitations of currently available therapy for AD. Additionally, we also mentioned the drawbacks of herbal medicines under the heading “Issues and challenges with herbal drugs”.

As per reviewer’s suggestion, we have incorporated the following lines in the conclusion section. These additions are highlighted in the revised manuscript.

Multicenter clinical trials should be performed to validate the efficacy and safety of these herbal medicines alone or as formulations for the treatment of AD. The present article reviewed the reported efficacy of herbal medicines against AD in experimental and clinical studies. 

Round 2

Reviewer 2 Report

See below my reaction to your responses. For easier orientation on the text, I kept the original remarks and responses and inserted my new comments in red color. Only topics I am not happy with are included. I appreciate the authors reflected all my other remarks.

One important issue – the authors did not mark red all their changes in the MS text, so I may have overlooked something.

Comment: The authors describe the current treatment options for AD. I suggest they also mention current guidelines.

Reply: Thank you for bringing up this important point, but we would like to reiterate the fact that the present article discusses the herbal treatment of AD whereas majority of the guidelines deals with the management of AD as per modern medicine.

Comment after revision: However, the authors still discuss limitations of the current treatment options and mention for example tacrin. I would expect them to discuss these limitations based on clinical state of the art approach.

Comment: Last paragraph (lines 96-98) reads: “Therefore, the present article reviews selected herbal drugs and formulations commonly used for memory related disorders. Some of the pre-clinical and clinical studies of plant extracts relevant to AD are mentioned in table 1.”

I have several remarks regarding that:

1 . “reviews selected herbal drugs” – selected how?

  1. “formulations commonly used” – where? How the authors selected the formulations?
  2. “Some of the pre-clinical and clinical studies…” – some? Why no all available or all relevant or all meeting certain standards?

These issues should be explained.

Reply: In the current review article, commonly used herbal drugs or formulations for the management of memory related disorders mentioned in Indian Traditional system of medicine have been reviewed.

Additionally, we reviewed all the pre-clinical and clinical studies of plant extracts relevant to AD in the present review article. To avoid confusion, we have removed the table in the revised manuscript.

Comment after revision: While I see the point of removing the table, I do not see any other change in the MS reflecting my comments. Also, the authors claim they focus only on AD, but apparently, they selected “formulations for the management of memory related disorders mentioned in Indian Traditional system of medicine”. Again, I recommend including a brief description of the “methods”. Also, the title and the claim of focusing on AD only is not really true.

Methods:

Comment: This section is completely missing. While it does not have to be a separate section (may be in the end of intro or in other suitable place), I believe, the authors should provide information about databases, key-words, selection and exclusion criteria, etc. Ideally the whole selection algorithm. Another issue is the focus on AD only instead of on all types of dementia. The authors report preclinical data, which in great part are not specific for AD. The clinical trials appear to be mostly AD related. This should be explained.

Reply: We thank the reviewer for raining important issues. As this is not a systematic review article, therefore, we have not used any inclusion or exclusion criteria for selection of the herbal drugs.

Additionally, we appreciate useful suggestion to include all type of dementia. However, it’s very difficult to cover all type dementia as dementia is very broad topic. Considering the length of article, we just focused on AD so that we could covered all relevant information about herbal drugs and AD. 

Comment after revision: The authors did not accept my comments. Above, the authors claimed they selected “formulations for the management of memory related disorders mentioned in Indian Traditional system of medicine”. This does not mean AD only. I do see the problem of specificity for AD, for that reason I recommended to broaden the subject.

Comment: 4.x sections (the description of the specific herbs) – this text is obviously going in great depth of the matter and I like it. I would just organize it in a more standard way, ideally into paragraphs appearing consistently in all sub-sections. For instance: plant description – active compounds – preclinical data – clinical data – toxicity (to mimic what the authors generally do, but not everywhere consistently). Please, unify the texts and if there is missing evidence, say so. Consider including a section describing basic pharmacologic properties (pharmacodynamics, pharmacokinetics…) and interaction potential if known. (I appreciate, that you touched the subject in the last section of the MS.) There is a nice review about ADME in herbal medicines (He at al. 2011, ADME Properties of Herbal Medicines in Humans: Evidence, Challenges and Strategies, doi: 10.2174/138161211795164194), just for inspiration.

Reply: We appreciate valuable suggestion made by reviewer. However, in present article, we mainly focused to review the beneficial effects of herbal medicines reported in preclinical and clinical studies. Therefore, we think, there is no need to include the sections such as plant description – active compounds, basic pharmacologic properties (pharmacodynamics, pharmacokinetics). Additionally, herb-drug interactions have been covered under the heading “Issues and challenges with herbal drugs”.

Comment after revision: The authors did not accept my comments and left the structure untouched. The plant descriptions are often in the text, but I still believe the text of the “monography” would benefit from a more standardized structure. The paragraph describing the herb-drug interactions is very general and does not include a single reference. Despite the authors day “However, only few such studies are available.” – I read this as “there are some relevant studies”.

Author Response

Reviewer 2

Comments and Suggestions for Authors

See below my reaction to your responses. For easier orientation on the text, I kept the original remarks and responses and inserted my new comments in red color. Only topics I am not happy with are included. I appreciate the authors reflected all my other remarks.

One important issue – the authors did not mark red all their changes in the MS text, so I may have overlooked something.

Reply: We thank reviewer for rigorously reviewing the manuscript and making useful suggestions. In the previous revision, we highlighted the additions in red color but may have missed highlighting some and are sorry for the inconvenience.

In the current revision, we have made changes in track change format and highlighted in red color.

Comment: The authors describe the current treatment options for AD. I suggest they also mention current guidelines.

Reply: Thank you for bringing up this important point, but we would like to reiterate the fact that the present article discusses the herbal treatment of AD whereas majority of the guidelines deals with the management of AD as per modern medicine.

Comment after revision: However, the authors still discuss limitations of the current treatment options and mention for example tacrin. I would expect them to discuss these limitations based on clinical state of the art approach.

Reply: In the revised manuscript, we have modified the limitations and as per reviewer’s suggestion, we have removed “tacrine” from the text. The addition can be seen from lines 66-73 and 89-93 in track change mode and highlighted in red color.

Comment: Last paragraph (lines 96-98) reads: “Therefore, the present article reviews selected herbal drugs and formulations commonly used for memory related disorders. Some of the pre-clinical and clinical studies of plant extracts relevant to AD are mentioned in table 1.”

I have several remarks regarding that:

1 . “reviews selected herbal drugs” – selected how?

  1. “formulations commonly used” – where? How the authors selected the formulations?
  2. “Some of the pre-clinical and clinical studies…” – some? Why no all available or all relevant or all meeting certain standards?

These issues should be explained.

Reply: In the current review article, commonly used herbal drugs or formulations for the management of memory related disorders mentioned in Indian Traditional system of medicine have been reviewed.

Additionally, we reviewed all the pre-clinical and clinical studies of plant extracts relevant to AD in the present review article. To avoid confusion, we have removed the table in the revised manuscript.

Comment after revision: While I see the point of removing the table, I do not see any other change in the MS reflecting my comments. Also, the authors claim they focus only on AD, but apparently, they selected “formulations for the management of memory related disorders mentioned in Indian Traditional system of medicine”. Again, I recommend including a brief description of the “methods”. Also, the title and the claim of focusing on AD only is not really true.

Reply: As per reviewer’s suggestion, we have incorporated the “Method section” in the revised manuscript. (lines 108 to 119) in track change mode, highlighted in red color.

In the title, we did not mention the term “memory related disorders”. We used the term “Alzheimer’s disease” as the article is focused on herbal medicine and Alzheimer’s disease.

The following statement “present article reviews selected herbal drugs and formulations commonly used for memory related disorders” has been modified. The revised statement reads as “present article reviews selected herbal drugs and formulations commonly studied for the treatment of Alzheimer’s disease”.

Methods:

Comment: This section is completely missing. While it does not have to be a separate section (may be in the end of intro or in other suitable place), I believe, the authors should provide information about databases, key-words, selection and exclusion criteria, etc. Ideally the whole selection algorithm. Another issue is the focus on AD only instead of on all types of dementia. The authors report preclinical data, which in great part are not specific for AD. The clinical trials appear to be mostly AD related. This should be explained.

Reply: We thank the reviewer for raining important issues. As this is not a systematic review article, therefore, we have not used any inclusion or exclusion criteria for selection of the herbal drugs.

Additionally, we appreciate useful suggestion to include all type of dementia. However, it’s very difficult to cover all type dementia as dementia is very broad topic. Considering the length of article, we just focused on AD so that we could covered all relevant information about herbal drugs and AD. 

Comment after revision: The authors did not accept my comments. Above, the authors claimed they selected “formulations for the management of memory related disorders mentioned in Indian Traditional system of medicine”. This does not mean AD only. I do see the problem of specificity for AD, for that reason I recommended to broaden the subject.

Reply: It is well known that Alzheimer’s disease (AD) is the major cause of dementia. Additionally, authors also have expertise in field of herbal medicine and AD. Considering the expertise of authors and burden of AD, in the present review article, we only focused on AD.

The following statement “present article reviews selected herbal drugs and formulations commonly used for memory related disorders” has been modified.  The revised statement reads as “present article reviews selected herbal drugs and formulations commonly studied for the treatment of Alzheimer’s disease”.

Comment: 4.x sections (the description of the specific herbs) – this text is obviously going in great depth of the matter and I like it. I would just organize it in a more standard way, ideally into paragraphs appearing consistently in all sub-sections. For instance: plant description – active compounds – preclinical data – clinical data – toxicity (to mimic what the authors generally do, but not everywhere consistently). Please, unify the texts and if there is missing evidence, say so. Consider including a section describing basic pharmacologic properties (pharmacodynamics, pharmacokinetics…) and interaction potential if known. (I appreciate, that you touched the subject in the last section of the MS.) There is a nice review about ADME in herbal medicines (He at al. 2011, ADME Properties of Herbal Medicines in Humans: Evidence, Challenges and Strategies, doi: 10.2174/138161211795164194), just for inspiration.

Reply: We appreciate valuable suggestion made by reviewer. However, in present article, we mainly focused to review the beneficial effects of herbal medicines reported in preclinical and clinical studies. Therefore, we think, there is no need to include the sections such as plant description – active compounds, basic pharmacologic properties (pharmacodynamics, pharmacokinetics). Additionally, herb-drug interactions have been covered under the heading “Issues and challenges with herbal drugs”.

Comment after revision: The authors did not accept my comments and left the structure untouched. The plant descriptions are often in the text, but I still believe the text of the “monography” would benefit from a more standardized structure. The paragraph describing the herb-drug interactions is very general and does not include a single reference. Despite the authors day “However, only few such studies are available.” – I read this as “there are some relevant studies”.

Reply: As per reviewer’s suggestions, we have included structure like plant description, main chemical constituents, pharmacological properties, preclinical studies, clinical studies if available and toxicity if available for each plant. The changes are in track change mode, highlighted in red color.

Regarding herb-drug interaction, we modified this section according to reviewer’s suggestion (lines 792-804) in track change mode, highlighted in red color.

Reviewer 3 Report

Unlike systematic review (SR), the Methods section is not mandatory for narrative review (NR), but if included, it adds clarity to the key messages. The quality of a narrative review may also be improved by borrowing from the systematic review methodologies that are aimed at reducing bias in the selection of articles for review and employing an effective bibliographic research strategy. In my opinion, an article without a methodology section is unacceptable and does not provide good enough information for publication.

Author Response

Comments and Suggestions for Authors

Unlike systematic review (SR), the Methods section is not mandatory for narrative review (NR), but if included, it adds clarity to the key messages. The quality of a narrative review may also be improved by borrowing from the systematic review methodologies that are aimed at reducing bias in the selection of articles for review and employing an effective bibliographic research strategy. In my opinion, an article without a methodology section is unacceptable and does not provide good enough information for publication.

Reply: As per reviewer’s suggestion, we have included “Method section” in the revised manuscript (line 108 to 119) in track change mode, highlighted in red color.
